# Knowledge, attitudes and preventive practices towards COVID-19 among workers at two points of entry in South-Eastern Botswana: A cross-sectional study

**Keatlaretse Siamisang**[1,2]*, **Naledi Mokgethi**[2], **Onalethata Lesetedi**[2], **Mpho Selemogo**[2]

1 Department of Family Medicine and Public Health, University of Botswana, Gaborone, Botswana,
2 Department of Health Services Management, Ministry of Health, Gaborone, Botswana

* drksiamisang@gmail.com

## Abstract

### Introduction

Adherence to control measures and provision of appropriate information at international borders and points of entry (POE) are key to limiting the importation of COVID-19. This study aimed to describe the knowledge, attitudes, and practices (KAPs) of POE staff towards COVID-19 in Botswana.

### Methods

This was a cross-sectional study of the COVID-19 KAPs among workers at Tlokweng border and Sir Seretse Khama International Airport (SSKIA) using a self-administered questionnaire. The tool incorporated the participants' demographics and selected questions on COVID-19 KAPs. Analysis was descriptive. Categorical data were summarized with frequencies while numeric data were summarized with medians and interquartile ranges (IQR). The total knowledge and practice scores of each individual were computed by adding their individual scores for each question. The scores were then categorized according to Bloom's cutoffs of good (80–100%), moderate (60–79%) and poor (<60%).

### Results

A total of 276 individuals participated in the study. Of these, 70 were from Tlokweng border and 206 were from SSKIA. The participants performed worst on questions on the frequency of severe disease and asymptomatic transmission of COVID-19. The attitudes were mainly positive. However, 54.6% of participants thought that the COVID-19 burden is exaggerated. For practice, the worst performance was on social distancing, sanitizing shared surfaces, and going to work while symptomatic. Overall, good and moderate knowledge was observed in 47.8% and 38.0% of participants, respectively. Similarly, good and moderate performance on practices was observed in 63.6% and 24.4% of participants respectively.

**Data Availability Statement:** All relevant data are within the paper and its Supporting Information files.

**Funding:** The authors received no specific funding for this work.

**Competing interests:** The authors have declared that no competing interests exist.

## Conclusion

The knowledge, attitudes, and practices were generally good at the 2 points of entry. More than 85% of respondents had moderate or good performance on knowledge and practice questions. However, the respondents performed poorly in some key questions. Targeted health information and promotion must address the identified gaps.

## Introduction

Corona Virus Disease 2019 (COVID-19) is caused by a novel corona virus known as Severe Acute Respiratory Syndrome Corona Virus-2 (SARS-Cov2) [1]. The World Health Organization (WHO) emergency committee convened under the International Health Regulations (IHR) and advised the director general to declare COVID-19 a Public Health Emergency of International Concern (PHEIC) [2, 3]. The declaration was made on 30 January 2020 [4, 5]. By definition, PHEIC is "an extraordinary event that may constitute a public health risk to other countries through international spread of the disease and may require an international coordinated response" [6, 7]. According to the International Health Regulations (IHR), airports, seaports, and land border crossings should have contingency plans and arrangements for dealing with PHEICs [8]. Indeed, through capacitating points of entry, international health emergencies can be controlled at their source [9].

In March 2020, the infection was declared a pandemic [1]. Several countries across the globe, including Botswana, have since enforced cross-border travel restrictions in response to the pandemic. While these restrictions are effective in controlling the infection spread, they are not sustainable in the long term. This is due to the negative economic impacts [10]. Countries were therefore encouraged to lift travel restrictions and implement other control measures. However, lifting or relaxation of travel restrictions can lead to importation or exportation of cases across borders. This can lead to new epidemics and can quickly reverse the gains made in controlling and containing outbreaks. New epidemics can force countries to return to drastic measures such as lockdowns [10]. This is undesirable due to the disruption of normal life, which has multiple social, economic, and mental health effects [11, 12]. Preventing importation of cases into the country can minimize these complications. Points of entry are therefore critical in controlling the importation and spread of COVID 19.

Botswana is a landlocked country in Southern Africa that shares major borders with South Africa, Namibia, Zimbabwe and Zambia. The country has not been spared by the COVID-19 pandemic [13, 14]. A significant number of cases in Botswana were imported. This was particularly true during the early stages of the pandemic [13]. Therefore, the need for the country to comply with the IHR guidance in this critical time cannot be overemphasized. Since the beginning of COVID-19 pandemic in 2020, several interventions were implemented at points of entry to build the capacity for staff to detect and prevent the introduction and spread of diseases in Botswana [15]. Workshops and seminars on infection prevention and control, case management, and contact tracing were conducted for the border and airport frontline workers. Health learning materials for both travelers and staff were developed and made readily available at the land border crossings and airports.

Adherence to control measures and provision of appropriate information at the points of entry are key to limiting the importation of COVID-19 into the country. This requires a collaborated and coordinated effort from all stakeholders at the POEs including port health staff, immigration, customs, police, and army officials. Individuals must adhere to COVID-19

protocols in their day-to-day duties and activities if importation of COVID-19 is to be prevented or minimized. Knowledge, attitudes, and practices (KAPs) are important predictors of this adherence [4, 11]. Points of entry staff are the first point of contact for returning travellers. They are therefore key players in controlling infection at the border. They also play a significant role in COVID-19 public information and education. Their KAPs can have a significant impact in the COVID-19 epidemic in the country. Currently, there are no studies on the KAPs towards COVID-19 among point of entry staff in Botswana. This must be addressed as KAP surveys can inform outbreak response including targeted health information [16]. They can also gauge the impact of health education and identify any gaps that may need urgent intervention. This study therefore, aimed to describe COVID-19 KAPs among workers at 2 points of entry in Botswana.

## Methods

### Study site and design

This was a cross-sectional study of the COVID-19 KAPs among workers at Tlokweng border and Sir Seretse Khama International Airport (SSKIA). SSKIA is located about 10 kilometers north of the capital city, Gaborone and is about 1 hour flight from Johannesburg, South Africa. It is the POE for regional and international travellers. It is the main and largest international airport in Botswana. It has the largest passenger movement in the country.

Tlokweng border is a ground crossing between Botswana and South Africa. The POE is in Tlokweng about 10 kilometers from Gaborone. It is one of the busiest ground crossings by passenger volume in Botswana.

### Study population

All health and non-health personnel at the 2 points of entry were eligible to participate in the study. This included but was not limited to immigration, customs, Botswana Police, Veterinary, and Port health staff.

### Selection of subjects and sample size calculation

Sampling was exhaustive and all consenting individuals were included in the study. All workers at the 2 points of entry were eligible for inclusion in the study regardless of their job cadre. Travellers, other clients and visitors were excluded. The investigators obtained written approval from the senior management at the points of entry. The research assistants then approached the workers in their work stations and invited them to participate in the study.

The formula for the minimum sample size in a cross sectional study is n = z*z P (1-P)/d*d where n is the sample size, z is the z statistic, P is the expected prevalence or proportion and d is the margin of error (13). The conventional z statistic of 1.96 for a 95% confidence interval was used and the precision was set at 5% or 0.05. In a similar study in Nigeria, 99.5% and 79.5% of respondents had good knowledge and good attitudes respectively [1]. A P of 0.795 was thus chosen. The calculated minimum sample size was 246.

### Data collection

The data was collected from 02 December 2021 to 11 March 2022. This coincided with the 4th (COVID-19 Omicron variant) wave. A self-administered questionnaire was used to collect data. The tool incorporated the participants' demographics and selected questions on COVID 19 KAPs. The research assistants gave the paper questionnaires to all consenting participants. The research assistants were available at both points of entry to address any questions during

the completion of the questionnaires and to collect the completed questionnaires. The participants were required to complete the questionnaire and return them to the research assistants on the same day. All efforts were made to check all returned questionnaires to ensure that they were appropriately and completely filled.

The KAP questions were adapted from previous literature [4]. Knowledge questions included transmission, common symptoms, vulnerable populations, and prevention and control measures. Attitudes were assessed through questions on attitudes towards COVID-19 preventive measures, while practice questions included social distancing, use of face masks, frequent washing of hands with soap and water, and use of sanitizers. There were 13 knowledge questions (Table 3 in S1 File) with 3 possible responses (yes, no, and I don't know). A correct answer was given a score of 1 while other answers got 0. There were 4 attitude questions with the same possible responses (Table 4 in S1 File). A Likert scale was used for the practice questions with 5 possible responses (never, rarely, sometimes, most of the time, always). The most appropriate answer was given a score of 4 while the least appropriate answer got 0.

### Data analysis

A data extraction sheet was used for all collected data. The data was entered into Microsoft Excel. After data cleaning and preparation, IBM Statistical Package for the Social Sciences (SPSS) version 26 was used for data analysis. Categorical data was summarized with frequencies. The numeric variable, age was skewed (Shapiro-Wilk test $p<0.001$). This variable was therefore summarized with a median and interquartile range (IQR). We report the participants' responses and performance on individual questions. The internal consistency of the knowledge and practices questions was computed. The Chronbach's alpha is reported.

The total knowledge and practice scores of each individual were computed by adding their individual scores for each question. The maximum knowledge score was 13 and the maximum practice score was 32 (8 questions x 4 maximum points). The knowledge and practices of individual participants were categorized according to the Bloom's cutoffs, where 80–100% represents good performance, 60–79% represent moderate performance, and $<60\%$ represents poor performance. This approach is consistent with previous COVID-19 KAP studies [17, 18].

### Ethical issues and protection of human subjects

Ethics approval was obtained from the University of Botswana Office of Research and development (ORD) and the Ministry of Health Research Unit. Permission to collect data was sought from all relevant authorities at the respective points of entry. Participation in the study was voluntary and informed consent was sought from all participants before they were enrolled in the study.

### Results

A total of 276 individuals agreed to participate in the study of which 70 were from Tlokweng border and 206 were from Sir Seretse Khama International Airport (SSKIA). The participants' characteristics are displayed in Table 1. The median age was 36 (IQR 29–46) at Tlokweng border and 34 (IQR 30–40) at SSKIA. Females accounted for 62.9% and 53.9% in Tlokweng and SSKIA, respectively. Only 16 (22.9%) and 50 (24.3%) participants at Tlokweng border and SSKIA, respectively, had no tertiary education. At Tlokweng border, 16 (24.6%) participants had been employed for less than 5 years, 9 (13.9%) had been employed for 6–10 years, 12 (18.5%) had been employed for 11-15years, 13 (20.0%) had been employed for 16–20 years and 15 (23.1%) had been employed for more than 20 years. At SSKIA, 115 (60.5%) had been employed for less than 5 years, 60 (31.6%), had been employed for 6–10 years, 10 (5.3%) had

**Table 1. Characteristics of participants.**

| Variable, n (%) | Tlokweng border (n = 70) | SSKIA (n = 206) |
|---|---|---|
| **Age** | | |
| *Data available* | *64 (91.4)* | *201 (97.6)* |
| **Median (IQR)** | 36 (29–46) | 34 (30–40) |
| **Range** | 21–62 | 20–55 |
| **Age category** | | |
| < 30 years | 18 (28.1) | 48 (23.8) |
| 30–39 years | 20 (31.2) | 103 (51.2) |
| 40–49 years | 16 (25.0) | 43 (21.4) |
| 50 years or more | 10 (15.6) | 7 (3.5) |
| **Sex** | | |
| *Data available* | *69 (98.6)* | *201 (97.6)* |
| Female | 44 (63.8) | 111 (55.2) |
| Male | 25 (36.2) | 90 (44.9) |
| **Education level** | | |
| *Data available* | *66 (94.3)* | *202 (98.1)* |
| No tertiary education | 16 (22.9) | 50 (24.3) |
| Diploma | 25 (35.7) | 74 (35.9) |
| Degree or higher | 25 (35.7) | 78 (37.9) |
| **Marital status** | | |
| *Data available* | *70 (100)* | *206 (100)* |
| Not married | 53 (75.7) | 146 (70.9) |
| Married | 17 (24.3) | 60 (29.1) |
| **Work (cadre)** | | |
| *Data available* | *69 (98.6)* | *192 (93.2)* |
| Customs | 8 (11.6) | 5 (2.6) |
| Immigration | 19 (27.5) | 8 (4.2) |
| Police | 15 (21.7) | 29 (15.1) |
| Port health | 15 (21.7) | 35 (18.2) |
| Military | 6 (8.7) | 6 (3.1) |
| Security | 0 (0.0) | 24 (12.5) |
| Aviation | 0 (0.0) | 10 (5.2) |
| Other * | 6 (8.7) | 75 (39.1) |
| **Length of employment** | | |
| *Data available* | *65 (92.9)* | *190 (92.2)* |
| 0–5 years | 16 (24.6) | 115 (60.5) |
| 6–10 years | 9 (13.9) | 60 (31.6) |
| 11–15 years | 12 (18.5) | 10 (5.3) |
| 16–20 years | 13 (20.0) | 1 (0.5) |
| >20 years | 15 (23.1) | 4 (2.1) |
| **Monthly income (BWP)** | | |
| *Data available* | *65 (92.9)* | *138 (67.0)* |
| <2000 | 1 (1.5) | 18 (13.0) |
| 2000–5000 | 14 (21.5) | 54 (39.1) |
| 5000–10000 | 28 (3.1) | 58 (42.0) |
| >10000 | 22 (33.9) | 8 (5.8) |

(*Continued*)

**Table 1.** (Continued)

| Variable, n (%) | Tlokweng border (n = 70) | SSKIA (n = 206) |
|---|---|---|
| **Age** | | |
| Not documented | 5 (7.1) | 68 (33.0) |

* Other: Airline staff, Engineer, Agriculture and Veterinary staff, Travel agents, Safety health and environmental officer, groundsmen, supplies officer

been employed for 11–15 years, 1(0.5%)_had been employed for 16–20 years while 4 (2.1%) had been employed for more than 20 years.

The medical history and living circumstances of the study participants are displayed in Table 2. Half of the participants at Tlokweng border reported flu-like symptoms in the past month compared to 26.6% in SSKIA. There were 18 (25.7%) participants with chronic medical conditions in Tlokweng border compared to 18 (8.7%) in SSKIA. Half of the participants at SSKIA reported ever testing positive for COVID-19 compared to 42.9% in Tlokweng border. Thirty (42.9%) of participants at Tlokweng border reported stigma related to their work compared to 17.8% in SSKIA.

The Chronbach's alpha was 0.66 for the knowledge questions and 0.83 for the practice questions. This shows good internal consistency of the data collection tool. The participants' responses and performance in the knowledge questions are displayed in Table 3. The participants performed best in knowledge questions 1, 10, and 8. Question 1 was about the clinical symptoms of COVID-19 and 96.0% of participants gave a correct answer. Question 10 was "to prevent infection by COVID-19, individuals should avoid going to crowded places" and 92.4% of participants gave a correct response. Question 8 was "Face masks are effective in reducing the spread of COVID-19" and 90.9% of participants gave a correct response. The worst performance was seen in questions 5, 2, and 7. Question 5 was "most people with COVID-19 develop severe disease" and only 50.5% of participants gave a correct response. Question 2 was "young people are more likely to develop severe disease than older people" and only 60.4% of respondents gave a correct response. Question 7 was "persons with COVID-19 cannot transmit the virus to others when they do not have fever" and only 61.1% of respondents gave a correct response.

**Table 2. Medical history and living circumstances of participants.**

| Variable, n (%) | Tlokweng border (n = 70) | SSKIA (n = 206) |
|---|---|---|
| Flu-like symptoms in the past month | 35 (50.0) | 54 (26.6) |
| Chronic medical conditions | 18 (25.7) | 18 (8.7) |
| Long term medications | 14 (20.0) | 18 (8.7) |
| History of asthma | 12 (17.1) | 18 (8.7) |
| History of heart disease | 4 (5.7) | 3 (1.5) |
| History of cancer | 5 (5.7) | 1 (0.5) |
| Ever tested positive for COVID-19 | 30 (42.9) | 103 (50) |
| Long term COVID-19 effects | 18 (25.7) | 20 (9.7) |
| Ever been in quarantine | 36 (51.4) | 129 (62.9) |
| Alcohol intake | 31 (44.3) | 64 (31.1) |
| Experience of stigma based on work | 30 (42.9) | 16 (7.8) |
| Elderly (>65years old) in household | 20 (28.6) | 42 (20.4) |
| People with chronic conditions in household | 16 (22.9) | 15 (7.3) |

**Table 3. Participants' responses to knowledge questions.**

| Knowledge items, n (%) | Data available | Correct | Incorrect | Do not know |
|---|---|---|---|---|
| 1. The main clinical symptoms of COVID-19 are fever, dry cough, and muscle pain. | 273 (98.9) | 262 (96.0) | 7 (2.7) | 4 (1.5) |
| 2. Young people are more likely to develop severe disease than older people | 273 (98.9) | 165 (60.4) | 82 (30.0) | 26 (9.5) |
| 3. Currently there is no cure for COVID-19 | 272 (98.6) | 223 (82.0) | 41 (15.1) | 8 (2.9) |
| 4. Most people with COVID-19 die | 266 (96.3) | 168 (63.2) | 85 (32.0) | 13 (4.7) |
| 5. Most people with COVID-19 develop severe disease | 275 (99.6) | 139 (50.5) | 116 (42.2) | 20 (7.2) |
| 6. People with underlying chronic illness are more likely to develop severe disease than the general population | 273 (98.9) | 206 (75.5) | 46 (16.8) | 21 (7.7) |
| 7. Persons with COVID-19 cannot transmit the virus to others when they do not have a fever | 270 (97.8) | 165 (61.1) | 83 (30.7) | 22 (8.1) |
| The COVID-19 virus spreads via respiratory droplets of infected individuals | 269 (97.5) | 239 (86.6) | 22 (8.0) | 8 (2.9) |
| 8. Face masks are effective in reducing the spread of COVID-19 | 275 (99.6) | 250 (90.9) | 20 (7.3) | 5 (1.8) |
| 9. It is not necessary for children and young adults to take measures to prevent the infection by the COVID-19 virus | 272 (98.6) | 196 (72.1) | 71 (26.1) | 5 (1.8) |
| 10. To prevent the infection by COVID-19, individuals should avoid going to crowded places | 276 (100) | 255 (92.4) | 12 (4.3) | 9 (3.3) |
| 11. Isolation of people who are infected with the COVID-19 virus is an effective way of reducing the spread of the virus. | 276 (100) | 246 (89.1) | 28 (10.1) | 2 (0.7) |
| 12. People who have contact with someone infected with the COVID-19 virus should be immediately isolated in a proper place. | 275 (99.6) | 245 (89.1) | 20 (7.3) | 10 (3.6) |

Table 4 shows the participants' responses to the attitude questions. A total of 219 (79.9%) of the respondents believed that COVID-19 would finally be successfully controlled while 225 (81.8%) had confidence that Botswana will win the battle against COVID-19. When asked if they thought they were at an increased risk of COVID-19 due to the nature of their job, 234 (85.7%) agreed. More than half (54.6%) of participants believed the COVID-19 burden is exaggerated to create fear and/or restrict people's freedoms.

The participants' responses to the practice questions are displayed in Table 5. In the preceding 2 weeks, 79.4% of participants had always worn a mask when going out, 45.4% had always managed to keep a distance of at least 1 metre from other people in public places, 60.1% had always managed to wash their hands or used a sanitizer after interacting with other people or touching shared objects and 68.1% had always worn a mask correctly (covering the nose and

**Table 4. Responses to attitude questions.**

| Attitude questions, n (%) | Data available | Yes | No | Don't know |
|---|---|---|---|---|
| 1.Do you believe that COVID-19 will finally be successfully controlled? | 274 (99.3) | 219 (79.9) | 44 (16.1) | 11 (4.0) |
| 2. Do you have confidence that Botswana will win the battle against COVID-19? | 275 (99.6) | 225 (81.8) | 36 (13.1) | 14 (5.1) |
| 3. Do you think that you are at a significantly increased risk of COVID-19 infection due to the nature of your job? | 273 (98.9) | 234 (85.7) | 35 (12.8) | 4 (1.5) |
| 4. Do you believe COVID-19 burden is exaggerated to create fear and/or restrict people's freedoms | 271 (98.2) | 148 (54.6) | 110 (40.6) | 13 (4.8) |

**Table 5. Responses to practice questions.**

| Practice questions, n (%) | Data available | Never | Rarely | Sometimes | Most of the time | Always |
|---|---|---|---|---|---|---|
| 1. In the past 2 weeks, how often have you worn a mask when going out? | 272 (98.6) | 13 (4.8) | 5 (1.8) | 22 (8.1) | 16 (5.9) | 216 (79.4) |
| 2. In the past 2 weeks, how often do you manage to keep a distance of at least 1metre from other people in public places? | 273 (98.9) | 8 (2.9) | 14 (5.1) | 36 (13.2) | 91 (33.3) | 124 (45.4) |
| 3. In the past 2 weeks, how often have you washed your hands or used a sanitizer after interaction with other people or touching shared objects? | 273 (98.9) | 12 (4.4) | 10 (3.7) | 22 (8.1) | 65 (23.8) | 164 (60.1) |
| 4. In the past 2 weeks, how often have you worn a mask correctly (covering nose and mouth) when interacting with colleagues at work? | 273 (98.9) | 8 (2.9) | 10 (3.7) | 20 (7.3) | 49 (17.9) | 186 (68.1) |
| 5. In the past 2 weeks, how often have you NOT managed to keep a distance of at least 1 meter between yourself and colleagues in the work place | 266 (96.3) | 25 (9.4) | 35 (13.2) | 84 (31.6) | 59 (22.2) | 63 (23.7) |
| 6. In the past 2 weeks, how often have you been able to sanitize shared surfaces and objects at least 3 times daily at work? | 268 (97.1) | 10 (3.7) | 19 (7.1) | 47 (17.5) | 82 (30.6) | 110 (41.0) |
| 7. In the past 2 weeks, how often have you had your temperature checked before reporting for work and interacting with your colleagues? | 271 (98.2) | 32 (11.8) | 25 (9.2) | 36 (13.3) | 53 (19.6) | 124 (45.8) |
| P8. In the past 4 weeks, how often have you come to work when you have had flu-like symptoms (cough, fever, runny nose, etc.) | 271 (98.2) | 127 (46.9) | 47 (17.3) | 25 (9.2) | 18 (6.6) | 54 (19.9) |

mouth) when interacting with colleagues at work. In the preceding 2 weeks, 23.7% of the participants had always not managed to keep a distance of at least 1 meter between themselves and colleagues in the workplace, 41.0% had always been able to sanitize shared surfaces and objects at least 3 times daily at work and 45.8% had always had their temperature checked before reporting for work and interacting with their colleagues. Finally, 19.9% of participants had always come to work when they had flu-like symptoms.

The participants' performance on the knowledge and practice questions was categorized into poor (0–59%), moderate (60–79%), and good (80–100%). This is displayed in Table 6 below. For the knowledge questions, good, moderate, and poor performance was seen in 47.8%, 38.0%, and 14.1% of participants, respectively. In Tlokweng border, 38.6% of participants demonstrated good knowledge, 30.0% demonstrated moderate knowledge, and 31.4% demonstrated poor knowledge. In SSKIA, 51.0% had good knowledge, 40.8% had moderate knowledge, and 8.3% had poor knowledge. Overall, 63.8%, 24.1%, and 12.1% of the participants demonstrated good, moderate, and poor performance on the practice questions respectively. In Tlokweng border, 32.3%, 33.9% and 33.9% had good, moderate and poor performance respectively. In SSKIA, 73.3%, 21.3% and 5.5% achieved good, moderate and poor performance respectively.

**Table 6. Participants' performance on knowledge and practices questions.**

| | Tlokweng Border | SSKIA* | Overall |
|---|---|---|---|
| **Performance on knowledge questions** | | | |
| Good | 27 (38.6) | 105 (51.0) | 132 (47.8) |
| Moderate | 21 (30.0) | 84 (40.9) | 105 (38.0) |
| Poor | 22 (31.4) | 17 (8.3) | 39 (14.1) |
| **Performance on practices questions** | | | |
| Good | 20 (32.3) | 148 (73.3) | 168 (63.6) |
| Moderate | 21 (33.9) | 43 (21.3) | 64 (24.2) |
| Poor | 21 (33.9) | 11 (5.5) | 32 (12.1) |

* SSKIA: Sir Seretse Khama International Airport

## Discussion

This is the first report of COVID-19 knowledge, attitudes, and practices in Botswana. The results were positive for most of the individual questions. More than 90% of participants recognized the most common symptoms of the disease and were aware that wearing facemasks and avoiding crowded places were effective in controlling the spread of infection. This was higher than in a similar Korean study where 49% of participants believed masks were effective [17]. However, the Korean study was done during the early days of the COVID-19 pandemic. At that time, COVID-19 knowledge was still evolving. The vast majority of respondents recognized that the infection is spread through respiratory droplets, that isolation of infected people is an effective intervention, and that there is currently no cure for COVID-19. The respondents also performed well in questions on high-risk populations and the need for precautions even by low-risk individuals. However, the respondents performed poorly in some key knowledge questions. Only about half of the participants disagreed with the statement that most people with COVID-19 develop severe disease. Furthermore, only about 60% of the respondents gave appropriate responses for questions on disease severity and age as well as absence of fever and infection transmission. Similarly, about 60% of the participants believed that most people with COVID-19 die. This is in contrast to a study in Kenya in which more than 90% of respondents recognized that people with COVID-19 could transmit the infection even when they do not have a fever. Most of their respondents also correctly recognized that most people with COVID-19 do not develop severe disease [19]. Similar to our study, most outpatient service visitors in Ethiopia did not recognize that COVID-19 patients can transmit the infection when they do not have a fever [20].

Our findings are significant as knowledge is a key factor in enhancing appropriate health behaviors [17]. Our study identifies specific areas that need targeted health information interventions. It is particularly important to educate the POE staff on COVID-19 transmission even in the absence of symptoms including fever. It is well established that asymptomatic transmission plays a significant role in COVID-19 outbreaks [21]. The belief that COVID-19 cannot be transmitted when there is no fever may compromise infection prevention and control practices in the workplace and in the community. Similarly health education and promotion should emphasize risk factors for adverse outcomes including advancing age. This would not only allow people to gauge their own risk but that of their families and friends as well. Other similarly designed studies have shown high COVID-19 knowledge [4, 17].

The attitudes were mainly positive with about 80% of respondents agreeing that COVID-19 will finally be successfully controlled and that Botswana will win the fight against the disease. In a Kenyan study, 81% of respondents agreed that COVID-19 will be successfully controlled [19]. Similar attitudes were reported in a Chinese study where about 90% of the respondents believed that COVID-19 will be successfully controlled [4]. However, The findings are different from what was reported in Nigeria where more than half of the respondents were not happy with the national response [1]. The contrast could be due to differences in the timing of the data collection. The Nigerian study was done in the early phases of the pandemic when there was a lot of uncertainty and confusion about COVID-19 and its optimal management. More than three quarters of respondents in our study believed that they were at an increased risk of COVID-19 due to the nature of their job. However, more than half of the respondents believed that the COVID-19 burden is exaggerated to create fear or restrict freedoms. There is a clear perceived susceptibility that should encourage positive health behavior. However, the belief that the burden is exaggerated is a cause for concern. Further studies are needed to explore this. Public health interventions are needed to address this perception. The belief that COVID-19 will be successfully controlled is similar to findings from a KAP in China during

the early stages of the pandemic. In this online survey of Chinese nationals, more than 90% of the participants agreed that COVID-19 would finally be successfully controlled [4].

The practice questions were about wearing of masks, social distancing, COVID-19 infection prevention and control and screening. About 80% of participants reported consistent wearing of masks in public places. However, a lower proportion reported this in the workplace. In a Kenyan study, 90% of respondents reported consistent wearing of masks in the workplace [19]. Significantly more than half of the respondents in our study had gone to work when they had flu-like symptoms. These findings demonstrate significant room for improvement in COVID-19 practices at these points of entry. More studies are needed to explain why such a high proportion of the POE staff go to work while they are symptomatic. This practice may be driven by pressure from employers and supervisors. The proportion of people who consistently wore masks or socially distanced was lower than what was reported by Zhong et al. In their study, the vast majority of people had good practices. More than 96.4% avoided crowded places while 98.0% wore masks [4].

The overall performance of the participants in all questions was good with over 85% of the participants having good or moderate knowledge. Similarly, a high proportion of participants had moderate or good performance on the practice questions. This is reassuring. However, performance on individual questions remains a concern. The overall performance is similar to what was reported in a study in Cameroon. In this cross-sectional survey of 1,006 respondents, 84% had good knowledge. Participants also scored high in practices of social distancing, hand washing and wearing of masks [18]. Our study findings are also similar to those of a Ugandan study. In this survey of KAP of healthcare workers, 83.9% had good knowledge and 78.4% had positive attitudes. However, only 37% had good practices [19]. In a Nigerian online survey, 99.5% of respondents had good knowledge of COVID-19 [1]. The very high knowledge was attributed to the selected study population. The survey included mainly young, educated people who had access to the internet. In contrast, about a quarter of respondents in our study had no tertiary education. Our study findings are generally better than what was reported in Bangladesh during the early stages of COVID-19. In this survey, 48.4% of participants had good knowledge, 62.3% had positive attitudes, while only 55.1% had good practices [22]. The poor performance in this study is probably due to the timing of their data collection relative to the COVID-19 pandemic. Similar poor knowledge, attitudes, and practices were observed in Ethiopia relatively early in the pandemic. In this hospital-based study in August 2020, about 27% of outpatients had poor knowledge while about 44% had poor practices [20].

### Limitations

This study has some limitations. The data was collected from only 2 points of entry. This limits the generalizability of the findings. Furthermore, it was not possible to get the total number of eligible participants in each POE. Attempts to get this information through the different managers were unsuccessful. As the KAPs were self-reported, the study may be prone to misclassification resulting from social desirability bias. Despite these limitations, this study provides important insights into the knowledge, attitudes, and practices of POE staff in Botswana. It identifies important targets for health communication, information, and promotion.

### Conclusion

The knowledge, attitudes, and practices were generally good at the 2 points of entry. More than 85% of respondents had moderate or good performance on knowledge and practice questions. Despite this overall good performance, respondents performed poorly in some key questions. Targeted health information and promotion must address these important gaps.

## Supporting information

**S1 Data.**
(XLSX)

**S1 File.**
(DOCX)

## Acknowledgments

We would like to thank the following individuals for their great work at collecting data for this project: Caroline Pinky Mfolwe, Mogomotsi Maphula, Cecilia Saborole, Pako Kootlogele, Gamodimo Barrel, Lesego Matome, Tlotlo Kolopo, Keoagile Amos Letshoo, Keaoboka Osupeng and Segolame Lebogang. We are also grateful to the port health team at the Ministry of Health and the points of entry as well as the management and staff at the 2 points of entry. Finally, we would like to thank our colleagues at the Ministry of Health and University of Botswana Public Health Unit for their support.

This work is dedicated to our dear colleague, brother and friend, Dr. Oduetse Ratshipa, who made significant contributions in the design of this study including critical review of the protocol. Sadly, Dr. Ratshipa passed on before the study took off. May his soul continue to rest in eternal peace.

## Author Contributions

**Conceptualization:** Keatlaretse Siamisang, Naledi Mokgethi, Mpho Selemogo.

**Data curation:** Keatlaretse Siamisang.

**Formal analysis:** Keatlaretse Siamisang.

**Investigation:** Keatlaretse Siamisang, Naledi Mokgethi, Mpho Selemogo.

**Methodology:** Keatlaretse Siamisang, Naledi Mokgethi.

**Project administration:** Keatlaretse Siamisang, Naledi Mokgethi, Onalethata Lesetedi.

**Supervision:** Keatlaretse Siamisang, Onalethata Lesetedi, Mpho Selemogo.

**Visualization:** Keatlaretse Siamisang.

**Writing – original draft:** Keatlaretse Siamisang.

**Writing – review & editing:** Naledi Mokgethi, Onalethata Lesetedi, Mpho Selemogo.

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
