## [Decision Letter · Decision Letter 0]

22 Jul 2022

PONE-D-22-16696Knowledge, attitudes and practices towards COVID-19 at two points of entry in South-Eastern BotswanaPLOS ONE

Dear Dr. SIAMISANG,

Thank you for submitting your manuscript to PLOS ONE. After careful consideration, we feel that it has merit but does not fully meet PLOS ONE’s publication criteria as it currently stands. Therefore, we invite you to submit a revised version of the manuscript that addresses the points raised during the review process.

ACADEMIC EDITOR:1. Please revise the manuscript per the reviewers' comments.

2. Please follow the journal guideline to prepare the manuscript.

We look forward to receiving your revised manuscript.

Kind regards,

Farzad Taghizadeh-Hesary

Academic Editor

PLOS ONE

Journal Requirements: 

Additional Editor Comments:

Reviewer 1: he short form KAP should be used throughout the article in place of Knowledge, attitude, and practice. The analysis should not include incomplete data (data not documented). The name of the statistical test to test the difference in KAP between two POE needs to be mentioned. The data collection and study periods are missing, which are essential to interpret the results. Overall the language and presentation is not standard and needs proofreading.

Reviewer 2: ummary of the research and overall impression

The manuscript presents an interesting and useful study relatively novel, of KAP towards COVID-19 among staff at ports of entry in Botswana. Addressing COVID -19 or indeed any pandemic at ports of entry is important in controlling its spread. The manuscript is technically sound, written in an intelligible fashion and in standard English. The statistical analysis has been performed appropriately and the data supports the conclusions. However, the following issues raised below should be addressed to enhance the quality.

Reviewer 3: The manuscript stands to contribute to the exist body of knowledge regarding COVID 19. However, the following revision will be required.

Title:

The author may need to rephrase the title to put in to context the subjects.

Introduction:

Land borders crossing may be more appropriate as against border cross on line 67 etc

The statement on line 85 should be revised, it should be "in" as against "In".

The objective of the study as stated on lines 121 -122 is ambiguous as subject of the study was not mentioned. Additionally, the introduction section is a bit lengthy, it may be revised down in word count to contextually focus on the subject matter.

Methods:

The authors should provide information how the sample size for this study was estimated including the assumptions used in its estimation.

Though the authors have stated the source of that data collection being adapted from a previous study. It is unsure how the reliability of the tool was assessed ( Cronbach alpha) as well as the other measures used for standardizing the tool. Furthermore, since self administration method was used, it will be good if the authors can provide the details of the filled questionnaire retrieval mechanism used and how they were able to mitigate against missing data.

Results

The authors have used median and IOR as the summary indices for age but it is not clear how the normality of this variable was assessed. If age of the participants is skewed there is need for it to be mentioned in methods section.

Footnote should be provided to denote what "others" stand for in table 1 for religion and work cadre.

The author had stated that this was a descriptive study and but a test of comparison of level of knowledge, as well as for practice was done while a composite bar chart was used, it is unclear which statistical test was used and its rationale. If this is desired to be included, then it should be done properly in a contingency tabular format and it should be be captured in the objective and appropriate details included in the

method under data analysis section.

Discussion

The discussion can be revised such that results are not recited while the implications of the findings to the practice of public health are emphasized.

Major areas of improvement.

Title

1. The authors should include ‘staff’ or any phrase that conveys the study population in the title and also the type of study (cross-sectional). They should consider replacing ‘practices’ with ‘preventive practices’ or ‘preventive measures’ for a better conveyance of intended meaning.

Introduction

2. The authors should consider citing the following article for some COVID-19 response measures undertaken in Botswana: link provided

https://doi.org/10.12688/wellcomeopenres.16070.3

Methods

3. More detail is needed in the ‘study population and selection of subjects’ section. For example, the total number of eligible staff at each site, any inclusion and exclusion criteria, how the participants recruited.

Discussion

4. In the section under knowledge and attitude, it would be preferable to include comparisons with other African countries and discuss. Some of these studies were mentioned in the practice section only

Minor Areas of Improvement

1. The manuscript would benefit from minor grammar and spelling editing. For instance line 101 spelling should be referred, line 103 should be patients not patient.

Reviewer 4:

1. The manuscript requires English copy-editing.

2. The main conclusions-stating "The knowledge, attitudes, and practices were generally good" doesn't adhere to the results. The study found that 47% of the participants had GOOD knowledge of COVID-19, which is not satisfactory.

3. Introduction: Please correct the full term of Covid-19. Its correct full term is coronavirus disease 2019. Besides, please use same acronym of Covid-19 along the manuscript. Now, some are "Covid-19" and some others are "Covid 19".

4. Introduction, Line 80-81: Please cite the following relevant article for the sentence "This is undesirable due to the disruption of normal life,

which has multiple social, economic, and mental health effects".

- https://www.ncbi.nlm.nih.gov/pmc/articles/PMC8184167/

5. Introduction-Lines 102-103: The following sentence is hard to understand. Please revise it.

"Batswana who tested positive were refereed for facility isolation and foreigners were repatriated back to their respective countries."

Reviewers' comments:

Reviewer's Responses to Questions

**Comments to the Author**

1. Is the manuscript technically sound, and do the data support the conclusions?

Reviewer #1: Partly

Reviewer #2: Yes

Reviewer #3: Partly

Reviewer #4: Yes

2. Has the statistical analysis been performed appropriately and rigorously? 

Reviewer #1: No

Reviewer #2: Yes

Reviewer #3: Yes

Reviewer #4: Yes

3. Have the authors made all data underlying the findings in their manuscript fully available?

Reviewer #1: No

Reviewer #2: Yes

Reviewer #3: Yes

Reviewer #4: Yes

4. Is the manuscript presented in an intelligible fashion and written in standard English?

Reviewer #1: No

Reviewer #2: Yes

Reviewer #3: Yes

Reviewer #4: No

5. Review Comments to the Author

Reviewer #1: The short form KAP should be used throughout the article in place of Knowledge, attitude, and practice. The analysis should not include incomplete data (data not documented). The name of the statistical test to test the difference in KAP between two POE needs to be mentioned. The data collection and study periods are missing, which are essential to interpret the results. Overall the language and presentation is not standard and needs proofreading.

Reviewer #2: Summary of the research and overall impression

The manuscript presents an interesting and useful study relatively novel, of KAP towards COVID-19 among staff at ports of entry in Botswana. Addressing COVID -19 or indeed any pandemic at ports of entry is important in controlling its spread. The manuscript is technically sound, written in an intelligible fashion and in standard English. The statistical analysis has been performed appropriately and the data supports the conclusions. However, the following issues raised below should be addressed to enhance the quality.

Major areas of improvement.

Title

1. The authors should include ‘staff’ or any phrase that conveys the study population in the title and also the type of study (cross-sectional). They should consider replacing ‘practices’ with ‘preventive practices’ or ‘preventive measures’ for a better conveyance of intended meaning.

Introduction

2. The authors should consider citing the following article for some COVID-19 response measures undertaken in Botswana: link provided

https://doi.org/10.12688/wellcomeopenres.16070.3

Methods

3. More detail is needed in the ‘study population and selection of subjects’ section. For example, the total number of eligible staff at each site, any inclusion and exclusion criteria, how the participants recruited.

Discussion

4. In the section under knowledge and attitude, it would be preferable to include comparisons with other African countries and discuss. Some of these studies were mentioned in the practice section only

Minor Areas of Improvement

1. The manuscript would benefit from minor grammar and spelling editing. For instance line 101 spelling should be referred, line 103 should be patients not patient.

Reviewer #3: The manuscript stands to contribute to the exist body of knowledge regarding COVID 19. However, the following revision will be required.

Title:

The author may need to rephrase the title to put in to context the subjects.

Introduction:

Land borders crossing may be more appropriate as against border cross on line 67 etc

The statement on line 85 should be revised, it should be "in" as against "In".

The objective of the study as stated on lines 121 -122 is ambiguous as subject of the study was not mentioned. Additionally, the introduction section is a bit lengthy, it may be revised down in word count to contextually focus on the subject matter.

Methods:

The authors should provide information how the sample size for this study was estimated including the assumptions used in its estimation.

Though the authors have stated the source of that data collection being adapted from a previous study. It is unsure how the reliability of the tool was assessed ( Cronbach alpha) as well as the other measures used for standardizing the tool. Furthermore, since self administration method was used, it will be good if the authors can provide the details of the filled questionnaire retrieval mechanism used and how they were able to mitigate against missing data.

Results

The authors have used median and IOR as the summary indices for age but it is not clear how the normality of this variable was assessed. If age of the participants is skewed there is need for it to be mentioned in methods section.

Footnote should be provided to denote what "others" stand for in table 1 for religion and work cadre.

The author had stated that this was a descriptive study and but a test of comparison of level of knowledge, as well as for practice was done while a composite bar chart was used, it is unclear which statistical test was used and its rationale. If this is desired to be included, then it should be done properly in a contingency tabular format and it should be be captured in the objective and appropriate details included in the

method under data analysis section.

Discussion

The discussion can be revised such that results are not recited while the implications of the findings to the practice of public health are emphasized.

Reviewer #4: Comments:

1. The manuscript requires English copy-editing.

2. The main conclusions-stating "The knowledge, attitudes, and practices were generally good" doesn't adhere to the results. The study found that 47% of the participants had GOOD knowledge of COVID-19, which is not satisfactory.

3. Introduction: Please correct the full term of Covid-19. Its correct full term is coronavirus disease 2019. Besides, please use same acronym of Covid-19 along the manuscript. Now, some are "Covid-19" and some others are "Covid 19".

4. Introduction, Line 80-81: Please cite the following relevant article for the sentence "This is undesirable due to the disruption of normal life,

which has multiple social, economic, and mental health effects".

- https://www.ncbi.nlm.nih.gov/pmc/articles/PMC8184167/

5. Introduction-Lines 102-103: The following sentence is hard to understand. Please revise it.

"Batswana who tested positive were refereed for facility isolation and foreigners were repatriated back to their respective countries."

6. PLOS authors have the option to publish the peer review history of their article (what does this mean?). If published, this will include your full peer review and any attached files.

Reviewer #1: No

Reviewer #2: **Yes: **Dr. Adaoha Pearl Agu

Reviewer #3: **Yes: **Tolulope Olumide Afolaranmi

Reviewer #4: No

---

## [Author Response · Author response to Decision Letter 0]

30 Aug 2022

Response to reviewers

Reviewer 1: 

The short form KAP should be used throughout the article in place of Knowledge, attitude, and practice. 

Response

The short form KAP has been used throughout the document 

The analysis should not include incomplete data (data not documented). 

Response

The descriptive analysis has been revised. For each variable, we now report the proportion of participants with available data. The analysis now excludes “not documented”

The name of the statistical test to test the difference in KAP between two POE needs to be mentioned. 

Response

To ensure the analysis is consistent with the objectives and the methodology, the test of comparison is no longer reported.

NB. The statistical test that was used initially was the chi square test

The data collection and study periods are missing, which are essential to interpret the results. 

Response

The data was collected from 02 December 2021 to 11 March 2022 during the fourth wave. This is now stated in the manuscript.

Overall the language and presentation is not standard and needs proofreading.

Results

Proofreading has been done.

Reviewer 2: summary of the research and overall impression

The manuscript presents an interesting and useful study relatively novel, of KAP towards COVID-19 among staff at ports of entry in Botswana. Addressing COVID -19 or indeed any pandemic at ports of entry is important in controlling its spread. The manuscript is technically sound, written in an intelligible fashion and in standard English. The statistical analysis has been performed appropriately and the data supports the conclusions. 

Thank you very much

Reviewer 3: The manuscript stands to contribute to the exist body of knowledge regarding COVID 19. 

Thank you very much

Title:

The author may need to rephrase the title to put in to context the subjects.

Response

The title has been rephrased. It now reads “Knowledge, attitudes and preventive practices towards COVID-19 among workers at two points of entry in South-Eastern Botswana: a cross-sectional study”

Introduction:

Land borders crossing may be more appropriate as against border cross on line 67 etc.

Response

“Ground crossing has been replaced with “land border crossing”

The statement on line 85 should be revised, it should be "in" as against "In".

Response

“In” has been replaced with “in”

The objective of the study as stated on lines 121 -122 is ambiguous as subject of the study was not mentioned. 

Response

The subjects of the study are now mentioned in the objective. The objective now reads, “This study therefore, aimed to describe COVID-19 knowledge, attitudes and practices among workers at 2 points of entry in Botswana”

Additionally, the introduction section is a bit lengthy; it may be revised down in word count to contextually focus on the subject matter.

Response

The introduction has been revised and made more succinct and focused on the subject matter.

Methods:

The authors should provide information how the sample size for this study was estimated including the assumptions used in its estimation.

Response

The minimum sample size calculation and its assumptions have been stated

“The formula for the minimum sample size in a cross sectional study is n= z*z P (1-P)/d*d where n is the sample size, z is the z statistic, P is the expected prevalence or proportion and d is the margin of error (13). The conventional z statistic of 1.96 for a 95% confidence interval was used and the precision was set at 5% or 0.05. A P of 0.5 was thus chosen to achieve a minimum sample size of 385 participants. This sample size calculation was based on the assumption that the proportions of good knowledge, attitudes and practices would fall between 10% and 90%. It also assumes a random sample.” 

Though the authors have stated the source of that data collection being adapted from a previous study. It is unsure how the reliability of the tool was assessed (Cronbach alpha) as well as the other measures used for standardizing the tool. 

Response

The tool’s internal consistency (knowledge and practice questions) was computed. The Chronbach’s alpha was 0.66 for the knowledge questions and 0.83 for the practice questions. 

Furthermore, since self administration method was used, it will be good if the authors can provide the details of the filled questionnaire retrieval mechanism used and how they were able to mitigate against missing data.

Response

This has been addressed in the paper and the following statement has been added:

“The research assistants gave the paper questionnaires to all consenting participants. The research assistants were available at both points of entry to address any questions during the completion of the questionnaires and to collect the completed questionnaires. The participants were required to complete the questionnaire and return them to the research assistants on the same day. All efforts were made to check all returned questionnaires to ensure that they were appropriately and completely filled.”

Results

The authors have used median and IOR as the summary indices for age but it is not clear how the normality of this variable was assessed. If age of the participants is skewed there is need for it to be mentioned in methods section.

Response

This has been addressed. This statement is now added to the analysis section

“The numeric variable, age was skewed (Shapiro-Wilk test p<0.001). This variable was therefore summarized with a median and interquartile range.”

Footnote should be provided to denote what "others" stand for in table 1 for religion and work cadre.

Response 

A footnote has been provided for work cadre. 

The religion variable has been has been removed as almost all people identified as Christian and most of the “others” had missing data.

The author had stated that this was a descriptive study and but a test of comparison of level of knowledge, as well as for practice was done while a composite bar chart was used, it is unclear which statistical test was used and its rationale. If this is desired to be included, then it should be done properly in a contingency tabular format and it should be captured in the objective and appropriate details included in the

method under data analysis section.

Response

This is indeed a descriptive study. We have replaced the bar charts with a table. The table simply displays the performance on the knowledge and practice questions at the 2 sites. To ensure the analysis is consistent with the objectives and the methodology, the test of comparison is no longer reported.

NB. The statistical test that was used initially was the chi square test

Discussion

The discussion can be revised such that results are not recited while the implications of the findings to the practice of public health are emphasized.

Response

The discussion has been revised. The results are summarized and the implications of the findings are emphasized.

Major areas of improvement.

Title

1. The authors should include ‘staff’ or any phrase that conveys the study population in the title and also the type of study (cross-sectional). They should consider replacing ‘practices’ with ‘preventive practices’ or ‘preventive measures’ for a better conveyance of intended meaning.

Response

This has been addressed. The title now reads “Knowledge, attitudes and preventive practices towards COVID-19 among workers at two points of entry in South-Eastern Botswana: a cross-sectional study”

Introduction

2. The authors should consider citing the following article for some COVID-19 response measures undertaken in Botswana: link provided

https://doi.org/10.12688/wellcomeopenres.16070.3

Response

The paper is indeed relevant and has been cited 

Methods

3. More detail is needed in the ‘study population and selection of subjects’ section. For example, the total number of eligible staff at each site, any inclusion and exclusion criteria, how the participants recruited.

Response

All consenting workers at the 2 points of entry were enrolled in the study. None were excluded. This has been stated in the methods section.

“All workers at the 2 points of entry were eligible for inclusion in the study regardless of their job cadre. Travellers, other clients and visitors were excluded. The investigators obtained written approval from the senior management at the points of entry. The research assistants then approached the workers in their workstations and invited them to participate in the study”. 

Efforts to obtain the number of eligible staff in each site were unsuccessful. This has been acknowledged in the study limitations. 

Discussion

4. In the section under knowledge and attitude, it would be preferable to include comparisons with other African countries and discuss. Some of these studies were mentioned in the practice section only.

Response

The discussion has been revised to include comparisons with other studies in the knowledge and attitude section. 

Minor Areas of Improvement

1. The manuscript would benefit from minor grammar and spelling editing. For instance line 101 spelling should be referred, line 103 should be patients not patient.

Language editing has been done

Reviewer 4:

1. The manuscript requires English copy-editing.

Language editing has been done

2. The main conclusions-stating "The knowledge, attitudes, and practices were generally good" doesn't adhere to the results. The study found that 47% of the participants had GOOD knowledge of COVID-19, which is not satisfactory.

Response 

The conclusions are based on the fact that more than 85% of participants achieved good or moderate performance on the knowledge and practice questions. We have added this statement in the conclusion to clarify this “More than 85% of respondents had moderate or good performance on knowledge and practice questions.”

3. Introduction: Please correct the full term of Covid-19. Its correct full term is coronavirus disease 2019. Besides, please use same acronym of Covid-19 along the manuscript. Now, some are "Covid-19" and some others are "Covid 19".

Response

This has been addressed. The same acronym of COVID-19 is now used throughout the manuscript. 

4. Introduction, Line 80-81: Please cite the following relevant article for the sentence "This is undesirable due to the disruption of normal life,

which has multiple social, economic, and mental health effects".

- https://www.ncbi.nlm.nih.gov/pmc/articles/PMC8184167/

Response

The paper has been cited.

5. Introduction-Lines 102-103: The following sentence is hard to understand. Please revise it.

"Batswana who tested positive were refereed for facility isolation and foreigners were repatriated back to their respective countries."

Response

This part of the introduction has been removed, as the section was made more succinct and focused.

---

## [Decision Letter · Decision Letter 1]

4 Oct 2022

PONE-D-22-16696R1Knowledge, attitudes and preventive practices towards COVID-19 among workers at two points of entry in South-Eastern Botswana: a cross-sectional studyPLOS ONE

Dear Dr. SIAMISANG,

Thank you for submitting your manuscript to PLOS ONE. After careful consideration, we feel that it has merit but does not fully meet PLOS ONE’s publication criteria as it currently stands. Therefore, we invite you to submit a revised version of the manuscript that addresses the points raised during the review process.

We look forward to receiving your revised manuscript.

Kind regards,

Farzad Taghizadeh-Hesary

Academic Editor

PLOS ONE

Journal Requirements:

Additional Editor Comments:

Please find the reviewer no. 3's comments and revise the manuscript.

Reviewers' comments:

Reviewer's Responses to Questions

**Comments to the Author**

1. If the authors have adequately addressed your comments raised in a previous round of review and you feel that this manuscript is now acceptable for publication, you may indicate that here to bypass the “Comments to the Author” section, enter your conflict of interest statement in the “Confidential to Editor” section, and submit your "Accept" recommendation.

Reviewer #2: All comments have been addressed

Reviewer #3: (No Response)

Reviewer #4: All comments have been addressed

2. Is the manuscript technically sound, and do the data support the conclusions?

Reviewer #2: Yes

Reviewer #3: Yes

Reviewer #4: (No Response)

3. Has the statistical analysis been performed appropriately and rigorously? 

Reviewer #2: Yes

Reviewer #3: (No Response)

Reviewer #4: (No Response)

4. Have the authors made all data underlying the findings in their manuscript fully available?

Reviewer #2: Yes

Reviewer #3: (No Response)

Reviewer #4: (No Response)

5. Is the manuscript presented in an intelligible fashion and written in standard English?

Reviewer #2: Yes

Reviewer #3: Yes

Reviewer #4: (No Response)

6. Review Comments to the Author

Reviewer #2: All the comments in the various sections- Title, Introduction, Methods, Discussion- have been addressed. The concerns on language editing have also been addressed.

Reviewer #3: The rationale for using 50% or 0.5 as P in the sample size estimation is not very clear. The authors may need to consider using P from a previous study. The assumption that the proportion of individuals with good KAPs would fall between 10% and 90% seems not well substantiated in the light of the fact that knowledge, attitudes and practice were assessed separately.

Reviewer #4: My comments are convincingly addressed and the manuscript quality is substantially improved. I have no more comments.

7. PLOS authors have the option to publish the peer review history of their article (what does this mean?). If published, this will include your full peer review and any attached files.

Reviewer #2: **Yes: **Dr. Adaoha Pearl AGU

Reviewer #3: **Yes: **Tolulope Olumide Afolaranmi

Reviewer #4: No

---

## [Author Response · Author response to Decision Letter 1]

16 Oct 2022

Response to reviewers

Reviewer #3: The rationale for using 50% or 0.5, as P in the sample size estimation is not very clear. The authors may need to consider using P from a previous study. The assumption that the proportion of individuals with good KAPs would fall between 10% and 90% seems not well substantiated in the light of the fact that knowledge, attitudes and practice were assessed separately.

Response

Thank you for this comment. We have now used a previous similar study in Nigeria where 99.5% and 79.5% of respondents had good knowledge and attitudes respectively. We used the 79.5% to calculate the minimum sample size which comes down to 246 participants. We agree that since knowledge, attitudes and practices were assessed separately, it would be difficult to predict that the KAP would fall between 10% and 90%. We have therefore removed that statement.

---

## [Editor Report · Decision Letter 2]

31 Oct 2022

Knowledge, attitudes and preventive practices towards COVID-19 among workers at two points of entry in South-Eastern Botswana: a cross-sectional study

PONE-D-22-16696R2

Dear Dr. SIAMISANG,

We’re pleased to inform you that your manuscript has been judged scientifically suitable for publication and will be formally accepted for publication once it meets all outstanding technical requirements.

Kind regards,

Farzad Taghizadeh-Hesary

Academic Editor

PLOS ONE

---

## [Editor Report · Acceptance letter]

2 Nov 2022

PONE-D-22-16696R2 

Knowledge, attitudes and preventive practices towards COVID-19 among workers at two points of entry in South-Eastern Botswana: a cross-sectional study 

Dear Dr. Siamisang:

I'm pleased to inform you that your manuscript has been deemed suitable for publication in PLOS ONE. Congratulations! Your manuscript is now with our production department. 

Kind regards, 

on behalf of

Dr. Farzad Taghizadeh-Hesary 

Academic Editor

PLOS ONE